# Radiomics Analysis on [^68^Ga]Ga-PSMA-11 PET and MRI-ADC for the Prediction of Prostate Cancer ISUP Grades: Preliminary Results of the BIOPSTAGE Trial

**DOI:** 10.3390/cancers14081888

**Published:** 2022-04-08

**Authors:** Giacomo Feliciani, Monica Celli, Fabio Ferroni, Enrico Menghi, Irene Azzali, Paola Caroli, Federica Matteucci, Domenico Barone, Giovanni Paganelli, Anna Sarnelli

**Affiliations:** 1Medical Physics Unit, IRCCS Istituto Romagnolo per lo Studio dei Tumori (IRST) “Dino Amadori”, 47014 Meldola, Italy; enrico.menghi@irst.emr.it (E.M.); anna.sarnelli@irst.emr.it (A.S.); 2Nuclear Medicine and Radiometabolic Unit, IRCCS Istituto Romagnolo per lo Studio dei Tumori (IRST) “Dino Amadori”, 47014 Meldola, Italy; monica.celli@irst.emr.it (M.C.); paola.caroli@irst.emr.it (P.C.); federica.matteucci@irst.emr.it (F.M.); giovanni.paganelli@irst.emr.it (G.P.); 3Radiology Unit, IRCCS Istituto Romagnolo per lo Studio dei Tumori (IRST) “Dino Amadori”, 47014 Meldola, Italy; fabio.ferroni@irst.emr.it (F.F.); domenico.barone@irst.emr.it (D.B.); 4Biostatistics and Clinical Trials Unit, IRCCS Istituto Romagnolo per lo Studio dei Tumori (IRST) “Dino Amadori”, 47014 Meldola, Italy; irene.azzali@irst.emr.it

**Keywords:** prostate cancer, retrospective studies, MRI-ADC scans, [^68^Ga]Ga-PSMA-11 PET, radiomics

## Abstract

**Simple Summary:**

Radiomics analysis is used on magnetic resonance imaging – apparent diffusion coefficient (MRI-ADC) maps and [68Ga]Ga-PSMA-11 PET uptake maps to assess unique tumor traits not visible to the naked eye and predict histology-proven ISUP grades in a cohort of 28 patients. Our study’s main goal is to report imaging features that can distinguish patients with low ISUP grades from those with higher grades (ISUP one+) by employing logistic regression statistical models based on MRI-ADC and 68Ga-PSMA data, as well as assess the features’ stability under small contouring variations. Our findings reveal that MRI-ADC and [^68^Ga]Ga-PSMA-11 PET imaging features-based models are equivalent and complementary for predicting low ISUP grade patients. These models can be employed in broader studies to confirm their ISUP grade prediction ability and eventually impact clinical workflow by reducing overdiagnosis of indolent, early-stage PCa.

**Abstract:**

Prostate cancer (PCa) risk categorization based on clinical/PSA testing results in a substantial number of men being overdiagnosed with indolent, early-stage PCa. Clinically non-significant PCa is characterized as the presence of ISUP grade one, where PCa is found in no more than two prostate biopsy cores.MRI-ADC and [^68^Ga]Ga-PSMA-11 PET have been proposed as tools to predict ISUP grade one patients and consequently reduce overdiagnosis. In this study, Radiomics analysis is applied to MRI-ADC and [^68^Ga]Ga-PSMA-11 PET maps to quantify tumor characteristics and predict histology-proven ISUP grades. ICC was applied with a threshold of 0.6 to assess the features’ stability with variations in contouring. Logistic regression predictive models based on imaging features were trained on 31 lesions to differentiate ISUP grade one patients from ISUP two+ patients. The best model based on [^68^Ga]Ga-PSMA-11 PET returned a prediction efficiency of 95% in the training phase and 100% in the test phase whereas the best model based on MRI-ADC had an efficiency of 100% in both phases. Employing both imaging modalities, prediction efficiency was 100% in the training phase and 93% in the test phase. Although our patient cohort was small, it was possible to assess that both imaging modalities add information to the prediction models and show promising results for further investigations.

## 1. Introduction

Prostate cancer (PCa) is the second most frequent cancer diagnosis made in men and the fifth leading cause of death worldwide, with an ever-increasing incidence [1].

The current clinical-/PSA-based risk stratification for PCa still leads to a large number of men being overdiagnosed with indolent, early-stage PCa that may require only active surveillance rather than immediate treatment with unjustified comorbidities. According to pertinent societal guidelines, clinically non-significant PCa (cns-PCa) is characterized as ISUP grade one, where PCa is found in no more than two prostate biopsy cores, each affected by less than 50% of its length, with a total PSA inferior to 10 ng/mL [2,3]. At the same time, systematic trans-rectal ultrasound-guided 12-core biopsies may fail to detect the most aggressive components of PCa and their real sizes, underestimating clinically-significant PCa (cs-PCa) in up to 30% of cases and delaying active treatments. A noninvasive determination of the real ISUP grade group would be of great help in informing biopsy targeting and treatment decisions [4,5,6]. In this scenario, there is an emerging need for non-invasive methods that better correlate with histology-proven ISUP grades.

Multiparametric magnetic resonance imaging (mp-MRI), combining T1-weighted and T2-weighted sequences with Diffusion-Weighted MRI and [^68^Ga]Ga-PSMA-11 PET, have proven to be good candidates to bridge this gap [7,8,9,10]. For instance, the PROMIS trial demonstrated that mp-MRI at triage might avoid unnecessary biopsies in 27% of cases and allow for an 18% increase in the detection of clinically significant cancers for TRUS biopsies guided by mp-MRI compared to standard TRUS biopsies [7]. In subsequent studies, Kasivisvanathan et al. [11] and Ahdoot et al. [8] found that MRI-targeted biopsies are superior to standard transrectal ultrasonography-guided biopsies in men at clinical risk of prostate cancer. Furthermore, quantitative parameters extracted from Apparent Diffusion Coefficient maps (calculated from DWI sequences of mp-MRI) showed a negative correlation with histology-proven ISUP grade (formerly Gleason score) [12,13,14]. However, the positive predictive value of mp-MRI is still poor, ranging from 20% to 68% [12], resulting in needless biopsies [11] and a need for improvement, particularly for individuals classified as intermediate risk. Despite these new findings, the analysis of the MRI-ADC maps’ histograms alone leaves many grey areas in the discrimination of low ISUP grade patients (one vs. two+), which is critical for treatment guidance, such as deciding between active surveillance, surgery, or radiotherapy according to NICE guidelines [15].

Prostate-specific membrane antigen (PSMA) is a type II transmembrane glycoprotein overexpressed on the surfaces of PCa epithelial cells. Higher degrees of PSMA overexpression are associated with higher aggressive biology (Gleason Score/ISUP grade group), luminal subtype, high androgen receptor activity, and higher serum PSA and is related to tumor progression and disease recurrence [16,17,18,19,20,21,22]. Pioneering studies evaluating the potential of [^68^Ga]Ga-PSMA-11 PET to detect intraprostatic tumour foci have documented proportionality between the intensity of the PSMA tumour uptake and ISUP grade group, the size of tumor foci, tumor growth pattern (infiltrative versus expansive), serum PSA, and higher D’Amico scores [23,24,25,26,27,28,29,30,31,32]. A recent meta-analysis carried out on 389 patients with clinical/biochemical suspicion of PCa documented for [^68^Ga]Ga-PSMA-11 PET an overall sensitivity and specificity of 97% and 66%, respectively. Despite, [^68^Ga]Ga-PSMA-11 PET returned a poor specificity similar to that of mpMRI and its negative likelihood ratio was found to be 0.05, leading to a 20-fold decrease in the odds of PCa being present in patients with negative PET findings. In addition, the diagnostic accuracy of [^68^Ga]Ga-PSMA-11 PET for detecting clinically significant PCa returned pooled sensitivity and negative likelihood ratios of 0.99 and 0.02, respectively, implying a potential role as a non-invasive risk stratifier [33].

Thus, PSMA-targeted PET imaging has been proposed in recent years to increase the diagnostic accuracy of mpMRI in defining the malignant potential of lesions detected and scored according to PIRADS version 2.1 (accessed on 7 April 2022). Studies evaluating the added value of [^68^Ga]Ga-PSMA-11 PET and mpMRI for detection of clinically-significant PCa documented a significantly increased diagnostic accuracy for the multimodality approach compared to individual modalities. PSMA uptake (SUV(bw)_max_) and DWI MRI (ADC_max_ and ADC_min_) were found to be distinct biomarkers able to differentiate between clinically significant PCa and normal prostatic tissue in naïve prostate cancer patients with a Gleason Score ≥7 [34,35]. In this study, texture analysis, which applies advanced mathematical functions to medical images, will be employed both in MRI-ADC maps and [^68^Ga]Ga-PSMA-11 uptake maps to quantify peculiar tumor characteristics not visible to the naked eye, in order to predict the histology-proven ISUP grade. Therefore, their application to MRI-ADC maps has been reported to be helpful in reducing grey areas in ISUP grade prediction [36], and employed together with [^68^Ga]Ga-PSMA-11 PET, they may show even more promising results. Despite this evidence, Radiomics features are strongly affected both by acquisition parameters and contouring methods [37,38,39].

The primary objective of our study is to report the features able to discriminate low ISUP grade patients from higher grade (ISUP 1+) patients, employing both MRI-ADC and [^68^Ga]Ga-PSMA-11 data to test the stability of the features under small contouring variations.

## 2. Materials and Methods

### 2.1. Patient Selection

We retrospectively analyzed a dataset of mp-MRI and [^68^Ga]Ga-PSMA-11 PET images from 28 patients with biopsy-proven prostate adenocarcinoma enrolled in our institutional prospective multi-cohort study BIOPSTAGE (EudraCT number: 2017-002651-28) between May 2018 and May 2020. In this prospective study, patients with high-risk prostate cancer are staged by pelvic mp-MRI and [^68^Ga]Ga-PSMA-11 PET prior to radical prostatectomy and pelvic lymph node dissection to rule out metastases and for correlation of pelvic imaging findings with axial step section histopathology analysis. Both mp-MRI and [^68^Ga]Ga-PSMA-11 PET scans were performed in patients fulfilling the following cohort-specific inclusion criteria:(a)patients 18 years of age or older, able to express informed consent for study participation and compliant with BIOPSTAGE on-protocol imaging;(b)biopsy evidence of prostate cancer with any of the following high-risk characteristics:
(1)clinical T stage ≥ T2c;(2)clinical stage N1;(3)ISUP grade group ≥ 4;(4)serum PSA > 20 ng/mL;
(c)biopsies performed at least 4 weeks prior to mp-MRI and [^68^Ga]Ga-PSMA-11 PET;(d)patients opting for radical prostatectomy and pelvic lymph node dissection.


Exclusion Criteria included:
(a)ongoing hormone therapy at the time of screening and within the previous six months;(b)previous pelvic radiation therapy;(c)any medical condition incompatible with MRI scanning or the administration of MRI contrast medium, or any condition that impairs the quality of pelvic MRI imaging;(d)history of allergic reactions attributed to compounds of similar chemical or biological composition to [^68^Ga]Ga-PSMA-11;(e)other known malignant neoplastic disease in the patient’s medical history with a disease-free interval of less than 5 years; chemotherapy or radiation therapy in the 4 weeks prior to study entry;(f)a history of other malignant neoplastic disease in the patient’s medical history with a disease-free interval of less than 5 years.

An outline of the workflow employed prior to statistical analysis to obtain the results described below is given in Figure 1.

### 2.2. MR Imaging Protocol and Lesion Contouring

Mp-MRI studies were acquired at our department with a 3 Tesla MR Scanner (Philips Ingenia 3.0T, Philips Healthcare, Best, the Netherlands) by using a Philips Sense Flex Medium surface coil.

The patient was placed in the scanner in the supine position, feet first. T1-weighted (T1w), T2-weighted (T2w), and ADC maps generated by axial diffusion-weighted imaging (DWI) sequences for prostate/small pelvis were acquired.

Four b-values were used (b100, b800, b1000, b2000) to provide more accurate ADC calculations. Echo time (TE) and repetition time (TR) were ≤90 ms and ≥3000 ms, respectively. The field of view (FOV) was 16–22 cm with an in-plane dimension of 2.5 mm. Slice thickness was set to 3 mm without a gap.

ADC maps were selected for radiomic analysis since they are the most informative for lesion detection, localization, and characterization, providing essential information on neoplastic tissue and anatomic detail [14]. Contouring of the 28 patient lesions was performed on the mp-MRI through Watson Elementary software (Watson Medical, Nijmegen, the Netherlands) by an expert radiologist employing T1w, T2w, and ADC maps for a total of 37 lesions contoured.

### 2.3. [^68^Ga]Ga-PSMA-11 PET/CT Imaging Protocol and Lesion Contouring

[^68^Ga]Ga-PSMA-11 was prepared according to national regulations and good radiopharmaceutical practices (GRP) as outlined in specific EANM guidelines [26]. All patients were intravenously injected with a mean activity of 159 MBq of [^68^Ga]Ga-PSMA-11 (activity range: 112–202 MBq) via an indwelling catheter in an antecubital vein according to patient weight. A whole-body PET/CT scan was performed 60–80 min after I.V. administration of [^68^Ga]Ga-PSMA-11, covering a volume from the skull vertex through the mid-thigh in 3D flow motion. Whole-body PET acquisitions were corrected for attenuation and scatter and adjusted for system sensitivity and providing parametric images in terms of Standardized Uptake Values (SUV_bw_: KBq found/gm tissue/KBq injected/gm body mass). The PET reconstruction matrix was 400 × 400 pixels (Hi-REZ processing), achieving an axial resolution of 2.5 mm and a slice thickness of 4 mm. The CT component of the studies was performed using the CARE Dose4D protocol for CT dose adaptation (mAs weighed on z-axis, patient’s dimensions, and x-y axis), HDFOV 512 × 512 matrix, and slice thickness of 3 mm for PET attenuation correction and co-registration. [^68^Ga]Ga-PSMA-11 PET images were contoured by an expert Nuclear Physician on MIM Maestro software, employing as the minimum positivity threshold an arbitrary maximum SUV(bw)_max_ of 3 g/mL and outlining 62 positive lesions.

### 2.4. Histopathological Reporting and ISUP Grade Assignment

Post-surgical histopathology results are considered the standard of truth for ISUP grade determinations of lesions. The anatomic pathology specimens were sectioned serially from apex to base and submitted as 12 whole-mount sections for examination. After detailed microscopic revision, the ISUP grade pattern present in each section was determined. Then each lesion detected with mp-MRI or [^68^Ga]Ga-PSMA-11 PET was compared with the histopathology results and consequently, associated with an ISUP grade. If an imaging-detected lesion had negative correspondence with histopathology, this same lesion was classified as a false positive and consequently discarded from the analysis. All the lesions with tumour correspondence with histopathology were classified as true positives.

### 2.5. Images Pre-Processing and Radiomic Analysis

[^68^Ga]Ga-PSMA-11 PET and MRI-ADC images were resampled to a resolution of 1/1/1 mm to uniform the dataset.

For the Radiomics features’ stability assessments, physicians’ contours were isotropically expanded by 1 and 2 mm and contracted by 1 mm. Contraction of 2 mm was not considered in the analysis due to the small size of many lesions that may cause failure in the subsequent Radiomics analyses.

The lesions subset visible both in MRI-ADC and [^68^Ga]Ga-PSMA-11 PET were further contoured according to the following rules: (a) if the [^68^Ga]Ga-PSMA-11 PET lesion contour is included, the MRI-ADC lesion contour—[^68^Ga]Ga-PSMA-11 PET contour is chosen and vice versa (b) In case of partial overlapping (>80% of the volume), intersection between lesion contours was performed (c) in other scenarios, association between lesions was not considered.

First, second, and higher order features were extracted with the Image Biomarker Standardisation Initiative (IBSI) [40] compliant tool, SOPHiA DDM^™^ For Radiomics (2021 SOPHIA GENETICS s.p.a., Boston, MA, USA), for MRI-ADC and [^68^Ga]Ga-PSMA-11 PET images. First order features were derived from the histogram of voxel intensities. Second and higher order features were calculated from Intensity size-zone, co-occurrence, and run length-based matrices. Detailed descriptions of the 218 imaging features extracted can be found in the IBSI Reference manual [40]. Grey level quantization was fixed to 32 bins between the minimum and maximum values inside the Region Of Interest (ROI). Features extracted from physicians’ contours were compared with isotropically expanded and contracted ROIs (+1 and +2 mm −1 mm) through the Intra Class Correlation coefficient (ICC) to select stable features under small variations in contouring with ICC >0.6.

### 2.6. Statistical Analysis

The endpoint of this study was an investigation of the diagnostic performance of Radiomics features extracted from multimodality imaging (MRI-ADC and [^68^Ga]Ga-PSMA-11 PET) against the ISUP grade obtained from histological evaluation, in particular, the ability of radiomic features to discriminate ISUP 1 from higher grades in order to help with treatment stratification. In Figure 2, the entire process of statistical analysis is summarized, from features extraction to final model evaluations.

Five independent predictive logistic models for ISUP Grade were developed based on:(a)lesions visible only through [^68^Ga]Ga-PSMA-11 PET;(b)lesions visible only with MRI-ADC;(c)lesions visible with [^68^Ga]Ga-PSMA-11 PET and MRI-ADC but only employing 68-[68Ga]Ga-PSMA-11 PET imaging features;(d)lesions visible with [^68^Ga]Ga-PSMA-11 PET and MRI-ADC but only employing mp-MRI imaging features;(e)lesions visible both with [^68^Ga]Ga-PSMA-11 PET and MRI-ADC, with features extracted from both imaging modalities.

The models were built through a stochastic cross-validation process to evaluate their performance.

The modeling process followed the following procedure:

Lesion feature datasets were divided into training (2/3) and test (1/3) sets. Subsequently, a logistic regression model was trained on the training set, with the employing features selected by a least absolute shrinkage and selection operator (LASSO) algorithm with internal 3-fold cross-validation. The predictive ability of the model was then calculated on the test set. This operation was repeated 30 times and subsequently, receiver operating curves (ROC) and the area under the curve (AUC) of each iteration were recorded both for the training and test sets.

The models’ quality was reported by averaging the AUC across iterations. The ROC and AUC were reported for the best-performing iteration to evaluate the model’s prediction power and to compare the performances of the mixed imaging features model (e) with standalone imaging models (c) and (d). The most frequently selected features across iterations were reported as the most informative features for ISUP Grade prediction.

All statistical analyses were carried out with R and the open-source software RStudio [41]. The raw data of this study ([^68^Ga]Ga-PSMA-11 PET, MRI-ADC and Pathology records) are available from the corresponding author on reasonable request.

## 3. Results

Patients were aged between 44 and 72 years (mean age: 62 years). The median total PSA at the time of prostate cancer diagnosis was 6.8ng/mL (IQR: 4.4–8,7). Eleven patients had ISUP grade one prostate cancer in post-prostatectomy pathology, eight patients had ISUP two, three patients had ISUP three, five patients had ISUP four, and one patient had ISUP five prostate cancer. The median time between [68Ga]Ga-PSMA-11 PET and mp-MRI was eight days, whereas the median time between advanced imaging and prostatectomy was 45 days. In post-prostatectomy pathology, organ-confined disease (pT2a to pT2c) was documented in 21 patients; seven patients were found with locally advanced disease (pTa to pT3b). Table 1 provides an overview of the patient’s features, while Appendix A provides a more detailed description of the patient’s characteristics.

In this cohort of high-risk prostate cancer candidates for surgery, MRI-ADC and [^68^Ga]Ga-PSMA-11 PET, yielded similar sensitivities (71.5% and 72.3%, respectively) and specificities of 99.5% and 90.5%, respectively, in detecting prostate cancer foci.

For the purpose of this study, we analyzed only true positive lesions on MRI-ADC imaging (*n* = 37) and [^68^Ga]Ga-PSMA-11 PET imaging (*n* = 49 lesions) that had positive correspondence with histopathology and that were used to build models (a) and (b). The small unbalance in the number of discovered lesions between imaging modalities is due to the fact that in four patients, multiple PET lesions had correspondence with only one big lesion in MRI-ADC maps and for three patients, MRI-ADC imaging was low quality or unreadable. Among these lesions, 31 were topographically paired at fusion and employed to build models (c), (d) and (e).

We extracted 218 imaging features with the Radiomics software Sofia (manufacturer, city, state (if USA), country) from MRI-ADC and [^68^Ga]Ga-PSMA-11 PET imaging. The extraction was performed on the original images and expanded lesion contours. Subsequently, ICC was applied with a threshold of 0.6 to assess the features’ stability for small variations in contouring. Twenty-nine and 87 features successfully passed the ICC test for [^68^Ga]Ga-PSMA-11 PET and MRI-ADC imaging, respectively. These features were further investigated and employed to build the five logistic models described in the “Materials and Methods” section. Table 2 summarizes the performance of the models in the training and test phases and the overall best performing model for each category, with details described below. For [^68^Ga]Ga-PSMA-11 PET features (a), the average model performance in terms of the area under the curve (AUC) in the training and test sets were 0.58 and 0.53, respectively. One iteration out of the 30 showed a very good predictive power with an AUC of 0.90 in the training set and 1.00 in the test set. MRI-ADC -based models (b) exhibited higher performance with an average AUC of 0.91 in the training phase and 0.67 in the test set. Furthermore, eight of the 30 iterations showed high predictive performance both in the training and test sets with an AUC higher than 0.80. The average performance of model (c) based on [^68^Ga]Ga-PSMA-11 PET features but trained on lesions visible also for mp-MRI, was 0.80 and 0.60 on the training and test sets, respectively. One iteration returned an AUC of 0.95 in the training set and an AUC of 1.00 in the test set. The most frequently selected features for the models’ development were area density, inverse elongation, zone size non-uniformity, flatness, and volume fraction differences between the intensity fractions.

The average performance of models (d) based only on MRI-ADC features and trained on commonly detected lesions was 0.74 and 0.45 on the training and test sets, respectively. Two iterations scored an AUC higher than 0.80 and the most-selected features were joint maximum, zone distance non-uniformity, 90th discretized intensity percentile, compactness, information correlation, and skewness. Models (e) based on both [^68^Ga]Ga-PSMA-11 PET and MP-MRI features showed a mean performance of 0.75 on the training set and 0.49 on the test set. Two iterations had an AUC higher than 0.80 and the most informative features were normalized inverse difference ([^68^Ga]Ga-PSMA-11 PET), zone distance non-uniformity (MRI-ADC), joint maximum (MRI-ADC), large zone low grey level emphasis ([^68^Ga]Ga-PSMA-11 PET), 90th discretized intensity percentile(MRI-ADC), and area density ([^68^Ga]Ga-PSMA-11 PET).

In Figure 3, we report the ROC curves of the best performing iterations of models (c), (d), and (e), both in the training and test sets.

## 4. Discussion

The biopsy’s ISUP grade differs from the final ISUP determined after surgery in around one-third of patients, with biopsies tending to underestimate cancer aggressiveness. The differences between the two ISUPs can have a big impact on how patients are managed. As a result, incorporating pre-therapeutic imaging characteristics to accurately determine PCa aggressiveness is of great clinical importance.

This study evaluated the ability of MRI-ADCand [68Ga]Ga-PSMA-11-based quantitative analyses to help differentiate low-risk prostate cancer patients (ISUP one) from the higher-risk patient classes (ISUP>1) and aimed to evaluate the benefits of the two imaging techniques combined. However, the results of this paper can be intended only as proof of concept as the number of concordant lesions on MRI-ADC and [68Ga]Ga-PSMA-11 PET is low, which represents the major limitation of this study. To overcome this limitation, we employed a stochastic cross-validation approach and ran the LASSO-logistic modelling process on 30 partitions of the datasets into training and test sets. Another limitation is represented by the laborious and time-consuming process required to contour, fuse, and evaluate lesions on different imaging modalities. Furthermore, Radiomics feature variability due to imaging acquisition and reconstruction is another disadvantage that to date limits the widespread clinical practice of this approach. The average predictive power in terms of the AUC for the training phase is very variable across models (a)–(e) and reaches a maximum of 0.91 for model (b). In the test phase, the performances are quite low, ranging from 0.45 in model (d) to 0.65 in model (b). From these average AUCs, it is difficult to speculate about the benefits of employing both [68Ga]Ga-PSMA-11 PET and MRI-ADC for ISUP predictions and these low performances can be justified by the small datasets and mild class imbalance involved in the analysis that may compromise the training of the majority of the models. For model (e), we had a performance drop in the test phase probably caused by the augmented number of features involved in the analysis together and the reduction in the number of lesions. Following these results, we are convinced that the models’ predictive power was strongly influenced by the data repartition in the training and test phases and thus it is our opinion that only in the higher AUC models were the datasets correctly balanced to give an idea of the real benefit of imaging features. For these reasons, we should take a closer look at every single model to give further details about the contribution of the two imaging modalities. The best-performing [68Ga]Ga-PSMA-11 PET models (a)–(c) have very high accuracies (>90%) both in the training and test phases and outperform the baseline single-modality models similarly reported by Solari et al. [42] for PCa ISUP grade prediction.

It is interesting to point out that in (b) models, the MRI-ADC mean value that is the imaging predictor currently employed in clinical practice to assess patient risk was not selected by LASSO. This evidence suggests that the Radiomics approach can provide a significant improvement to patient classification for MRI-ADC sequences. Furthermore, the performances of the best training models (b) are in line with the previously reported performances of mp-MRI Radiomics-based analyses [43], in particular in the work of Fehr et al. [44]. By combining ADC and T2w mean values with textural features, they achieve an accuracy higher than 90% in differentiating low Gleason (6) prostate lesions from higher scores (>7).

Building the predictive models (c) and (d), including the lesions visible both in MRI–ADC and [^68^Ga]Ga-PSMA-11 PET, we can assess that the two imaging modalities are equivalent in discriminating the low-risk patients from the higher-risk ones with an AUC of the best performing iteration of 1.00 for the test phase, as visible from Figure 3C,D. Finally, combining the 29 features of [^68^Ga]Ga-PSMA-11 PET and the 87 features of MRI–ADC imaging, we obtained model (e), where performances were slightly lower and had a maximum AUC of 0.93 (Figure 3F). It is important to point out that in this model, the LASSO algorithm always chooses the most informative of both MRI-ADC- and [^68^Ga]Ga-PSMA-11-based features to build the logistic regression prediction model, indicating that the two modalities contribute to adding unique information for lesion classification. However, with our dataset it is difficult to observe statistically significant improvements in the performances given by the integration of the two modalities due to the restricted number of lesions, and further investigation is required to confirm our hypothesis.

## 5. Conclusions

Among the developed models, each imaging modality seems to provide similar results in ISUP grade prediction. Preliminary results suggest that aside from the MRI-ADC average value, currently employed in clinical practice to assess lesion severity, other imaging biomarkers may provide complementary information for ISUP grade prediction, but further, broader studies are necessary to confirm these findings.

Both [^68^Ga]Ga-PSMA-11 PET and MRI-ADC imaging biomarkers showed to be complementary in ISUP grade assessment when employed together to build prediction models.

## Figures and Tables

**Figure 1 cancers-14-01888-f001:**
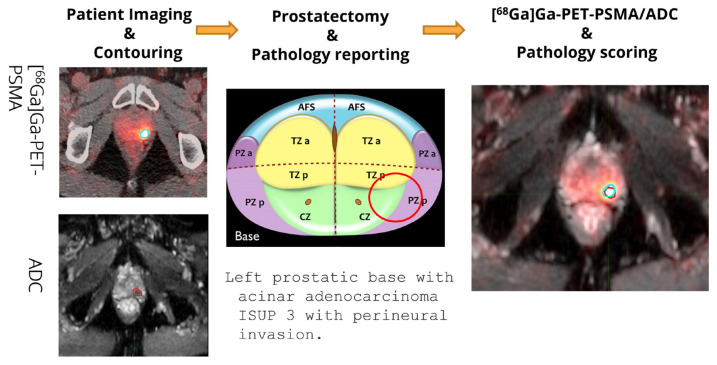
Left: Detail of prostate contouring for the two imaging modalities performed by Nuclear Physician and Radiologist, respectively. Center: Representative example of anatomic pathology reporting with details about ISUP grading. Right: [^68^Ga]Ga-PSMA-11 PET and MRI-ADC are fused with MIM maestro software with the respective contouring superimposed.

**Figure 2 cancers-14-01888-f002:**
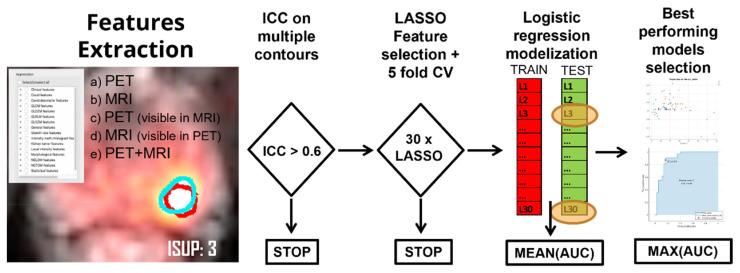
Detail of the workflow employed, from features extraction to selection of the final statistical models.

**Figure 3 cancers-14-01888-f003:**
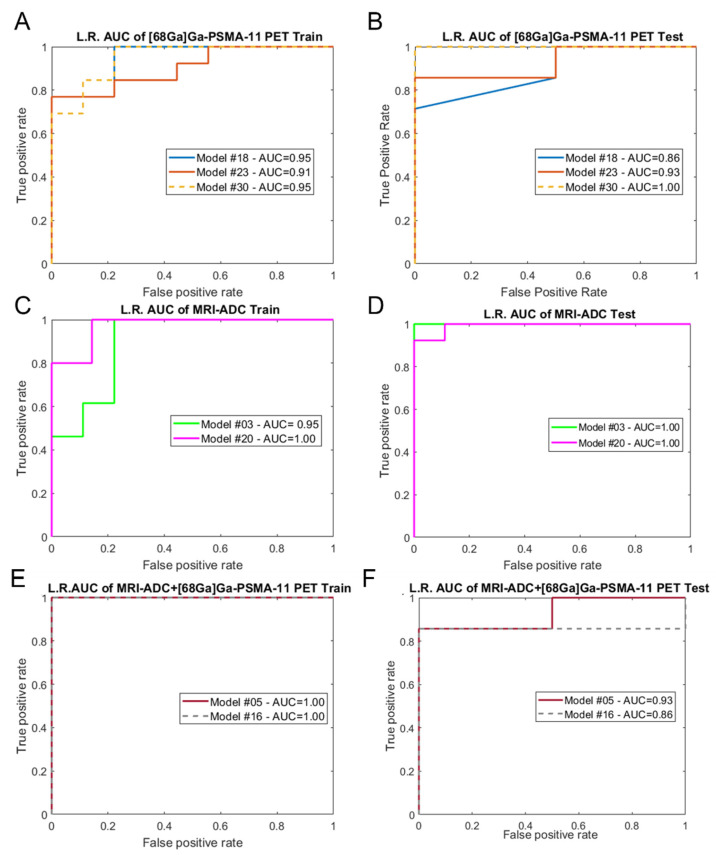
Logistic regression (L.R.) models’ performance in the training and test phases in terms of the area under the curve (AUC). In (**A**,**B**) AUCs of the best performing models (c) in the train and test phase are displayed. The same way in (**C**,**D**) the AUCs of models trained on features extracted from MRI-ADC are shown. Finally in (**E**,**F**) AUCs of models built combining [^68^Ga]Ga-PSMA-11 PET and MRI-ADC features are shown.

**Table 1 cancers-14-01888-t001:** Summary of patients’ characteristics.

Patients Characteristics	Value
mean age (years), age range	62.0 (44–72)
median age (years), IQR	63.0 (58.5–66.5)
median total PSA (ng/mL), IQR	6.8 (4.4–8.7)
median PSA density (ng/mL/g), IQR	0.15 (0.11–0.23)
median prostate volume (mL), IQR	48 (37.3–59.3)
overall ISUP grade group (post-prostatectomy pathology)	
1	*n* = 11
2	*n* = 8
3	*n* = 3
4	*n* = 5
5	*n* = 1
pathology T stage	
T2a-T2b	*n* = 6
T2c	*n* = 15
T3a	*n* = 4
T3b	*n* = 3
median time between [^68^Ga]Ga-PSMA-11 PET and mpMRI (days), IQR	8 (4–13)
median time between imaging and surgery (days), IQR	45 (24–86)

IQR: Inter Quartile Range, PSA: Prostate-Specific Antigen.

**Table 2 cancers-14-01888-t002:** Summary of trained and tested imaging biomarker-based models.

Model Type	Number of Lesions	Train Mean AUC	Test Mean AUC	Train Best AUC	Test Best AUC
PET	49	0.58	0.53	0.9	1
MRI	37	0.91	0.67	0.92	1
PET (MRI-visible)	31	0.8	0.6	0.95	1
MRI (PET-visible)	31	0.74	0.45	1	1
MRI+PET	31	0.75	0.49	1	0.93

## Data Availability

All raw data employed for the development of logistic regression models are available upon reasonable request to the corresponding author.

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
