# Peer review of "Radiomics Analysis on [68Ga]Ga-PSMA-11 PET and MRI-ADC for the Prediction of Prostate Cancer ISUP Grades: Preliminary Results of the BIOPSTAGE Trial"

_cancers, 2022, doi:10.3390/cancers14081888_

Round 1

Reviewer 1 Report

The authors show the radio mics analysis on 68-Ga-PSMA PET and MRI-ADC for the prediction of prostate cancer ISUP grades. The study is interesting, however, I have some concerns to be discussed.

  1. I would say that the numbers are not enough to confirm the conclusion or claim of the current study.
  2. How many teslas is the MRI? 1.5 or 3.0?
  3. The Figures are too small to see clearly. Please enlarge those.
  4. Do you use the SUV-max value in the current study?
  5. Please discuss disadvantages of the current methods.

Author Response

Hereafter, “C# stands for Reviewer’s comments and “R#” for our replies. All the reference numbers (pages, citations, etc.) refer to the “new” version of the text, unless explicitly otherwise defined. In the answers, the references to manuscript’ sentences are reported between “< … >”. All changes to the text are reported in red.

COMMENT OF REVIEWER#1:

[C1.1]: I would say that the numbers are not enough to confirm the conclusion or claim of the current study.

[R1.1]: We thank the reviewer for the comment, we removed the phrase “MRI-ADC average value, currently employed in clinical practice to assess lesion severity seems to be less relevant than other imaging biomarkers” and modified it as follows: “Preliminary results suggests that aside of MRI-ADC average value, currently employed in clinical practice to assess lesion severity, other imaging biomarkers may provide complementary information for ISUP grade prediction but further broader studies are necessary to confirm these findings.”

[C1.2]: How many teslas is the MRI? 1.5 or 3.0?

[R1.2]: We wrote in the methods section line 173-174:

“Mp-MRI studies were acquired at our department on a 3 Tesla MR Scanner (Philips  Ingenia 3.0T, Philips Healthcare, Best, Netherlands) by using a Philips Sense Flex Medium surface coil.”

[C1.3]: The Figures are too small to see clearly. Please enlarge those.

[R1.3]: We thank the reviewer for pointing this out, we enlarged the figures as requested

[C1.4]: Do you use the SUV-max value in the current study?

[R1.4]: Yes, we use SUVmax as the minimum positivity threshold as we wrote in line 203-206:

“[68Ga]Ga-PSMA-11 PET images were contoured by an expert Nuclear Physician on MIM-maestro software employing as minimum positivity threshold an arbitrary maximum SUVbw of 3 g/ml and outlining 62 positive lesions”.

Furthermore, SUVmax was employed in all the analyses described in the paper as a quantitative feature.

[C1.5]: Please discuss disadvantages of the current methods.

[R1.5]:  We thank the reviewer for the comment, we expanded the discussion section including the following phrase to argument the disadvantages of the approach:

“Another limitation is represented by the laborious and time-consuming process required to contour, fuse and evaluate lesions on different imaging modalities. Furthermore, Radiomics feature variability due to imaging acquisition and reconstruction is another disadvantage that to date limits the widespread in clinical practice of this approach.”

Reviewer 2 Report

cancers-1609937

The study by Feliciani et al. aims to test the use of MRI-ADC and PSMA PET to distinguish low and higher ISUP grades in a series, using a radiomic approach. Obtained findings show that MRI-ADC and 68-Ga-PSMA PET imaging features based models are equivalent and complementary in predicting low ISUP grades.

Given the unmet clinical need to non-invasively identify the real ISUP grade of PCa patients, the topic of the study is of high clinical interest. The paper is well structured, clearly written, and obtained results are rigorously and critically analyzed.

Just a few minor considerations from my side:

  • In the current version of the manuscript, radiopharmaceutical names are not reported according to the official nomenclature and somewhere are unclear (i.e., “68-GaPET” in Table 2). The authors are encouraged to check the radiopharmaceutical’s names according to the EANM Guidance to Radiotracer Nomenclature https://www.eanm.org/publications/guidelines/nomenclature/.
  • Several results are included in the methods section of the manuscript (i.e., lines 151-160 and Table 1). The authors should consider moving these parts to the study results.
  • The BIOPSTAGE trial recruited high-risk PCa patients undergoing pelvic mp-MRI and PSMA PET before radical prostatectomy and pelvic lymph node dissection. Surprisingly, 11/28 (39.2%) high-risk PCa patients included in the present study showed ISUP grade 1 at post-surgical histopathology. The authors should add a supplementary table detailing clinical T stage, clinical stage N1, ISUP grade and serum PSA of enrolled patients.
  • The authors should improve the discussion of the manuscript. They should discuss obtained findings given the existing literature on the topic. They should also further enhance the potential clinical impact of the study.  

Author Response

Hereafter, “C# stands for Reviewer’s comments and “R#” for our replies. All the reference numbers (pages, citations, etc.) refer to the “new” version of the text, unless explicitly otherwise defined. In the answers, the references to manuscript’ sentences are reported between “< … >”. All changes to the text are reported in red.

COMMENT OF REVIEWER#2:

[C2.1]: In the current version of the manuscript, radiopharmaceutical names are not reported according to the official nomenclature and somewhere are unclear (i.e., “68-GaPET” in Table 2). The authors are encouraged to check the radiopharmaceutical’s names according to the EANM Guidance to Radiotracer Nomenclature https://www.eanm.org/publications/guidelines/nomenclature/.

[R2.1]: We thank the reviewer for pointing this out; we modified the nomenclature to meet the official names

[C2.2]: Several results are included in the methods section of the manuscript (i.e., lines 151-160 and Table 1). The authors should consider moving these parts to the study results.

[R2.2]: We thank the reviewer for the comment. We moved the description of patients to the results section

[C2.3]: The BIOPSTAGE trial recruited high-risk PCa patients undergoing pelvic mp-MRI and PSMA PET before radical prostatectomy and pelvic lymph node dissection. Surprisingly, 11/28 (39.2%) high-risk PCa patients included in the present study showed ISUP grade 1 at post-surgical histopathology. The authors should add a supplementary table detailing clinical T stage, clinical stage N1, ISUP grade and serum PSA of enrolled patients.

[R2.3]: We thank the reviewer for the comment. As requested, we added a supplementary table in the results section detailing the patients’ characteristics that justify the high risk PCa

[C2.4]: The authors should improve the discussion of the manuscript. They should discuss obtained findings given the existing literature on the topic. They should also further enhance the potential clinical impact of the study.  

[R2.4]:  We thank the reviewer for pointing this out, we improved the discussion session explaining further the clinical impact and considering most recent literature on the topic.

Round 2

Reviewer 1 Report

The authors replied well, so the manuscript is suitable for publication.